# High-Throughput Synthesis and Characterization Screening of Mg-Cu-Y Metallic Glass

**Dan J. Thoma** *[ID], **Janine T. Spethson, Carter S. Francis, Paul M. Voyles** [ID] **and John H. Perepezko**

Department of Materials Science and Engineering, the University of Wisconsin-Madison, 1509 University Ave, Madison, WI 53706, USA; csfrancis@wisc.edu (C.S.F.); paul.voyles@wisc.edu (P.M.V.); perepezk@wisc.edu (J.H.P.)
* Correspondence: dthoma@wisc.edu

**Abstract:** Bulk metallic glasses can exhibit novel material properties for engineering scale components, but the experimental discovery of new alloy compositions is time intensive and thwarts the rate of discovery. This study presents an experimental, high-throughput methodology to increase the speed of discovery for potential bulk metallic glass alloys. A well-documented system, Mg-Cu-Y, was used as a model system. A laser additive manufacturing technique, directed energy deposition, was used for the in situ alloying of elemental powders to synthesize discrete compositions in the ternary system. The laser processing technique can supply the necessary cooling rates of $10^3$–$10^4$ $Ks^{-1}$ for bulk metallic glass formation. The in situ alloying enables the rapid synthesis of compositional libraries with larger sample sizes and discrete compositions than are provided by combinatorial thin films. Approximately 1000 discrete compositions can be synthesized in a day. Surface smoothness, as discerned by optical reflectivity, can suggest glass-forming alloys. X-ray diffraction coupled with energy dispersive X-ray spectroscopy can further refine amorphous alloy signatures and compositions. Transmission electron microscopy confirms amorphous samples. The tiered rate of amorphous alloy synthesis and characterization can survey a large compositional space and permits a glass-forming range to be identified within one week, making the process at least three orders of magnitude faster than other discrete composition techniques such as arc-melting or melt-spinning.

**Keywords:** metallic glasses; high-throughput synthesis; laser processing; alloy design





## 1. Introduction

Metallic glasses have received sustained interest since their discovery in 1960 [1]. However, the discovery of new glass-forming systems in bulk samples is often limited by the fabrication rate of appropriate samples, and high-throughput screening methods are needed that could accelerate the discovery of new compositions. This current study focuses on an accelerated discovery method for Mg bulk metallic glasses. Attractive properties such as high specific strength and excellent corrosion resistance have made Mg-based metallic glasses (MGs) of special interest as a replacement for their crystalline counterparts, particularly in biomedical applications [2,3]. Metallic glasses typically require liquid cooling rates that bypass crystallization, thus often requiring limitations in a sample dimension in order to achieve rapid heat extraction. In contrast, bulk metallic glasses (BMGs) are alloys where the cooling rate to bypass the crystalline solid are sufficiently slow as to permit larger samples to be produced for engineering applications.

Experimentally, BMGs have been categorized by an ability to form an amorphous rod of 1 mm diameter or greater [4]. Typically, these rods are produced by suction arc casting. The materials discovery of BMGs often centers on searching and analyzing alloy space for compositions with good glass-forming ability (GFA). The GFA is the measure of how slowly the liquid structure can be transformed into an amorphous solid. It is inversely related to the critical cooling rate $R_C$, or the slowest rate at which the liquid can be cooled without crystallization. Since $R_C$ is difficult to measure directly, GFA is more commonly

determined using analogues like the critical casting thickness $Z_C$ or the critical diameter $D_C$ [5]. Data are limited for these measures of GFA as only glasses with $R_C < 10^3$ K s$^{-1}$ have a critical size large enough ($Z_C \geq 0.5$ mm, $D_C \geq 1$ mm) to reasonably synthesize and measure bulk properties [6–8]. However, the critical cooling rate lacks the necessary predictive power to find new composition glasses. Also, casting 1 mm diameter rods for searching compositional space is time consuming and may only produce tens of different compositions per week using conventional arc casting.

To increase the rate of BMG discovery, computational methods based on thermodynamic or kinetic models are often used to describe and predict glass formation in alloy systems [9,10]. Machine learning techniques are also being utilized [11]. Many of the most popular criteria are based on calculations from characteristic temperatures. However, these temperatures are determined experimentally using differential scanning calorimetry (DSC) [4,11–22]. Dozens of criteria have been proposed [10], but the most notable of these are the reduced glass transition temperature $T_{rg}$, which is the ratio between the liquidus $T_l$ and glass transition $T_g$ temperatures ($T_{rg} = T_l/T_g$) [11]; the supercooled liquid temperature range, that is the difference between $T_g$ and the onset of the first crystallization peak, $T_x$ ($\Delta T_x = T_x - T_g$) [4]; and the parameter $\gamma = T_g/(T_x + T_l)$ [12]. These criteria typically have fitting parameters from existing experimental data, so as with critical cooling rate, the predictive power is very poor for new systems. In other words, they may be able to identify the best glass-forming region in a known glass-forming alloy, but cannot predict the GFA in a new alloy system. Other parameters based entirely on known properties like atomic size, enthalpy and viscosity still need large amounts of experimental data for evaluation [15,16,23–25]. Since the physics of glass formation is not completely understood, the need for more experimental data is a necessity for metallic glass research. Therefore, new experimental methods with sufficient cooling rates are required for building larger databases with reduced experimental time for glass-forming alloys.

*High-Throughput Methods for MG Synthesis*

High-throughput experimentation (HTE) techniques have found increasingly greater use and impact, particularly in pharmaceuticals and catalysis research [26–28]. For materials research in polymers [29], oxide glasses [30], and high entropy alloys [31], high-throughput experimentation is largely centered around combinatorial efforts [12,29–37]. Combinatorial methods offer an attractive option to map alloy composition space for MGs, particularly given the sensitivity of GFA to composition [38–42]. For example, combinatorial film deposition can produce a very thin layer of material which contains a chemical gradient, where a single composition only exists at a small point [32,33]. In addition, effective cooling rates for the magnetron sputtering of these films are well above $10^6$ K s$^{-1}$ as the alloy is quenched from vapor [43–45]. This approach has shown success with Mg-based metallic glasses [32]. However, the small volume does not provide much material for characterization and may not be an appropriate analogue for the synthesis of bulk materials, so other synthesis routes are needed to produce larger scale metallic glass alloys.

Additive manufacturing (AM) permits the fabrication of amorphous products [46,47]. In addition, the technique allows for the high-throughput combinatorial fabrication of bulk production of samples with different compositions [31]. The AM of Mg alloys has largely been limited to laser powder bed fusion (LPBF) methods [48,49], which require pre-alloyed powder that does not readily allow for in situ alloying. However, in a directed energy deposition (DED) system, crystalline metallic powders are mixed in transit and melted with a laser in a positive pressure environment with <10 ppm $O_2$. DED processing results in quenches from the liquid at rates of $10^3$–$10^4$ K s$^{-1}$ [50–52], which is between the cooling rates of melt spinning and suction casting of rods and is sufficient to identify BMGs.

Most previous studies on combinatorial synthesis using laser processing from elemental powders have been focused on the functional grading of alloy systems [35–37,53–58]. The functional gradient studies are designed to analyze multiple compositions and are typically produced by changing composition as the part is built up, creating a vertical

gradient on a two-dimensional (2D) or three-dimensional (3D) sample. Typically, gradients are produced by fixing the total speed setting for the auger which delivers the powder from the hoppers (or less commonly the total mass flowrate) and changing the ratio of two powders.

Although functional grading with DED has proven to be effective in surveying compositional space and amorphous alloy formation, discreet alloy samples would be more practical to study metallic glass formation. Also, the ability to explore BMG Mg alloy space has not been explored with DED. Mg alloy metallic glasses are often based upon Mg-TM-RE (TM: transition metal; RE: rare earth) components [59]. These alloys present challenges in synthesis primarily due to disparate melting temperatures and the exceptionally high reactivity associated with Mg and the RE's. They also differ in powder flowability, which is closely related to the shape and size of the powder [60]. For this study, the Mg-Cu-Y system was chosen as a baseline as it is a well-known bulk-glass-forming system and follows the general Mg-TM-RE pattern.

In this experimental study, a laser alloy processing methodology is evaluated to investigate the 1–2 mm diameter volumes of samples with different alloy compositions. Discrete bulk samples allow for the use of X-ray diffraction (XRD). The synthesis method includes the use of four powder hoppers which enable the simultaneous deposition of up to four different powders without the need for additional printing. The compositional libraries produced by this method are characterized by a tiered high-throughput characterization method. The proposed methodology is being applied on a known glass-forming system which can act as a foundation for future systems to discover new bulk amorphous systems.

## 2. Materials and Methods

### 2.1. Materials

Elemental (>99.5% purity) Mg, Cu, and Y powders were obtained at nominally +100/−325 mesh size (40–150 μm diameter). The micrographs in Figure 1 show the size and shape of these powders as compared with 316L stainless steel (SS316L) powder, a common DED feedstock material. The Mg powder is smooth and oblong, which is typical of gas-atomized Mg powder, and is skewed towards the upper end of the recommended size distribution. The Cu powder is highly spherical, smooth, and free of satellite particles. Compared with the mostly smooth and spherical SS316L powder, the Cu and Mg powders have excellent size and shape for this AM process. The Y powder is blocky, with sharp edges, large aspect ratios, and a large fraction of particles below the lower size recommendation (85% of particles < 45 μm diameter). The substrate for all prints was a 99.9% pure Mg plate. To prevent back-reflection of the laser and subsequent damage to the optical laser system, the plate surface was roughened from the as-rolled state.

The printed samples were made using an Optomec LENS™ MR-7, a blown-powder directed energy deposition (DED) system. In this process (Figure 2), argon gas fluidizes powder from up to four individually controlled hoppers, mixes the powder en route to the print head, and delivers it through four nozzles in the print head which surround a central 1kW Nd:YAG laser with a 600 μm spot size. The chamber and powder hoppers were kept under a slight positive pressure of Ar (+0.007 atm) and a low $O_2$ content (<10 ppm) while printing.

### 2.2. High-Throughput Processing Methodology

With the LENS™, control over the printing path and processing parameters (like laser power and powder flowrate) occurs with programming changes to the G-code. To simplify the parameter space and minimize printing time, each composition was printed as a single bead using a pulse of the laser, followed by a single re-melting step, again using a single pulse of the laser. The initial deposition step used different laser power depending on the composition (240–330 W) but the re-melting step parameters were kept the same (285 W) across alloying space for consistency. The remelting step served to provide a more consistent cooling rate of the surface across samples regardless of the deposition

parameters, as well as to melt, smooth and homogenize the surface from any spurious powder welded to it from the initial deposition.

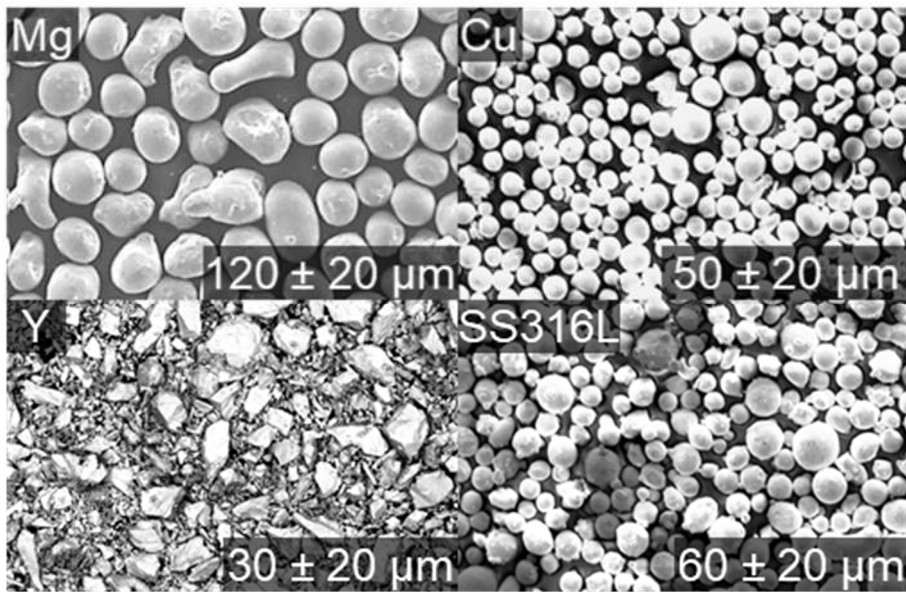

**Figure 1.** SEM images of the Mg, Cu, and Y powders used in this study compared with SS316L powder used for the same system. The recommendation for the system is a smooth, spherical powder with a 45–150 μm diameter; the SS316L powder recommended for the system is mostly smooth and spherical and has an average particle size of $60 \pm 20$ μm. Mg is rounded and skewed towards the upper end with an average particle size of $120 \pm 20$ μm. Cu is shaped ideally with a smooth, highly spherical powder at $50 \pm 20$ μm. Y is jagged and uneven, with large aspect ratios and a wide size distribution at an average of $30 \pm 20$ μm. As can be inferred from this comparison, Mg and Cu flow smoothly and consistently while Y requires an additional agitation of the powder to promote smooth flow.

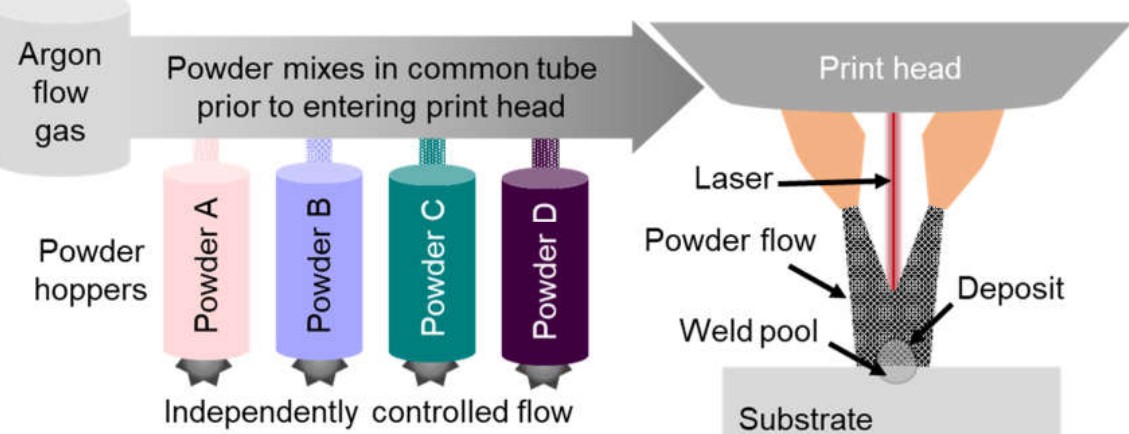

**Figure 2.** The LENSTM system uses up to four powders fluidized by inert gas at independently controlled flowrates. The powders mix in transit to the print head, where it is blown at a focal point on the substrate common to a 1 kW Nd:YAG laser. The area hit by the powder is approximately 2 mm in diameter, and the spot size for the laser is 600 μm. The laser heats the substrate to establish a melt pool that the powder enters and melts, building up the deposit to about 1 mm in diameter. Using different powder ratios and machine command language G-code, compositional libraries can be printed automatically.

### 2.3. Characterization

The tiered characterization method for this study is outlined in Figure 3. Following the production of a high volume of samples across an array of compositions (~1000 alloy compositions), an optical reflectivity method was used to screen out low quality and rough samples by an order of magnitude. The reduced number of samples were then analyzed with the more time-consuming X-ray diffraction (XRD) method, which further screens out unambiguously crystalline samples. The SEM-EDS measurements on select alloys evaluated actual compositions and deviations from the nominal alloy composition. Affirmation of an amorphous phase was accomplished using transmission electron microscopy (TEM).

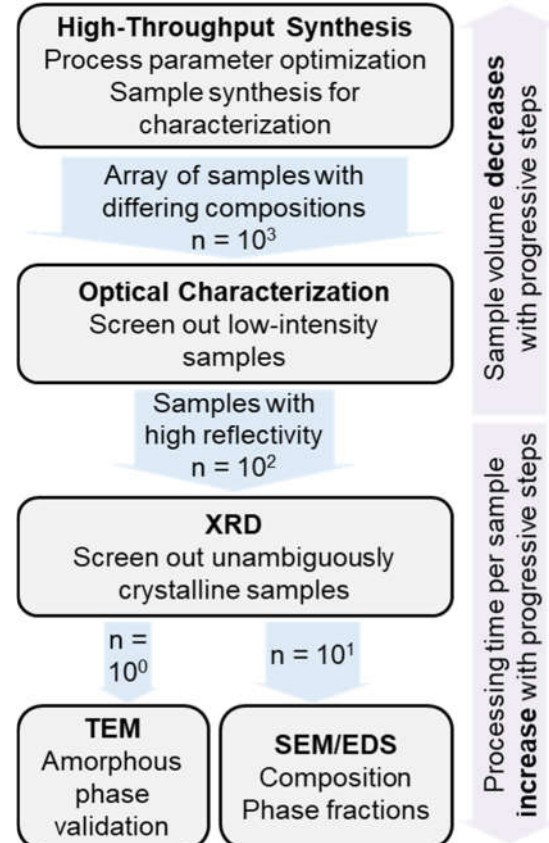

**Figure 3.** The tiered characterization method designed for this study was designed so the number of samples decreases as the time needed to analyze each sample increases. High-throughput synthesis was used to produce large quantities of samples across an array of compositions. Parameter-optimized samples are passed through to the optical screening method which removes samples with low print quality, rough surfaces or other clear signs that no amorphous material exists. The rest of the samples move on to XRD, which further removes samples which show spectra composed entirely of sharp crystalline peaks. The few samples remaining following this step have an amorphous phase positively confirmed with TEM, and composition and phase identification is carried out with SEM.

For optical reflectivity analyses, printed plates were imaged using diffuse lighting and a high-contrast background so images could be calibrated and compared across plates. A purely crystalline metal has small facets from preferred growth directions in randomly oriented grains, which scatter light and give a less reflective surface. The surfaces of amorphous metals are smooth and reflective. This is because the surface is free from the small facets that crystalline grains produce and the frozen-in liquid structure maintains the surface tension, giving a highly reflective polished look to the surface. This optical method was not used to find amorphous samples for further investigation, but rather to find clearly crystalline or low-quality samples to preclude from further investigation.

X-ray diffraction (XRD) was used on highly reflective beads to search for the peak broadening that is characteristic of an amorphous phase. The XRD was carried out on the surfaces of the beads using a Bruker D8 Discovery with Cu Kα radiation (0.3 mm aperture, 30 s dwell time) at 10° steps from an incident angle of 10–80°. Scanning electron microscopy (SEM) was performed using a Zeiss LEO 1530 Gemini equipped with electron dispersive spectroscopy (EDS) (15 kV accelerating voltage). Selected samples which showed some evidence of a broad amorphous scattering peak in XRD were further analyzed by TEM for validation. TEM validation was considered as the only definitive confirmation of an amorphous phase. Specimens for TEM were taken from the surface of the bead using focused ion beam (FIB) and analyzed with an FEI Tecnai T12 TEM also equipped with EDS (120 kV accelerating voltage).

## 3. Results

### 3.1. High-Throughput Synthesis of Nominal Alloy Compositions

Alloys in the entire Mg-Cu-Y ternary system, including binary systems, were printed on a Mg base plate as shown in Figure 4. The nominal compositions of the alloys were printed to be consistent with an isothermal section of a ternary phase diagram. A total of 595 beads were printed at 3 at% intervals in a few hours. More than one set of the 595 beads was processed within a day, thus permitting 1000 samples per day. The Mg-rich corner up to around 15 at% Y printed as relatively tall beads (taller than they were wide). The diameter of the beads was 1–2 mm. The beads became shorter as the nominal Y content rose. The beads in the Cu-rich corner were detached from the plate due to insufficient adhesion; these samples were eliminated from further analysis. The dark background of the base plate was metal condensate, which results from vaporization of metal in the molten pool that condenses on the plate as it cools. While the processing parameters were designed to minimize this effect, vaporization of the Mg metal was unavoidable at the input energies required to melt Y.

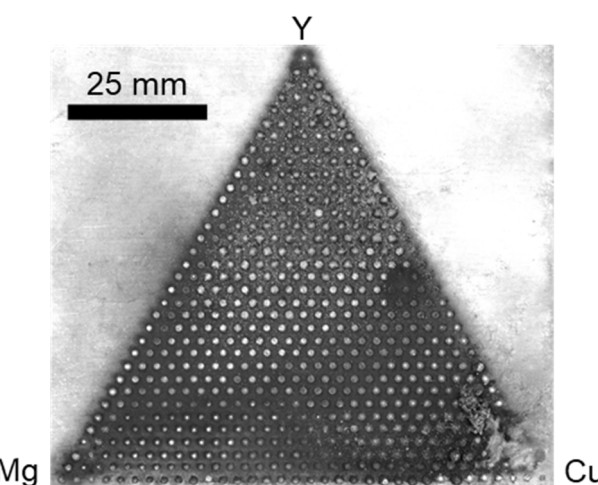

**Figure 4.** An optical image of a printed Mg-Cu-Y composition library taken with a high-contrast background and diffuse lighting. The nominal compositions of each bead range from 0–100 at% with 3 at% intervals in all three elements. This plate was printed using a 330 W laser power and a pulse duration less than 3 s, which varied according to composition. The total volume flowrate was kept constant at 2.4 cm$^3$ min$^{-1}$.

### 3.2. High-Throughput Optical Investigation of Amorphous Regions

The Mg substrate had normalized luminance values around 0.3 in the as-received state. Values of around 0.5 or higher were screened as potentially amorphous and selected for the next tier of characterization (XRD). The optical investigation process of identification took a few minutes.

Since Mg-Cu-Y was chosen as a well-documented MG forming system, the optical luminance measurements of the nominal compositions are compared to the literature results for glass-forming compositions [61–68] in Figure 5. In the literature results, compositions which have been experimentally identified as amorphous (mostly using the melt-spinning technique), crystalline, or partially amorphous are marked at the appropriate composition, and the critical casting diameter is additionally represented as a contour map to show the bulk-glass-forming region. The high-intensity regions of the optical measurements center around the same area as the known bulk-glass-forming region, although there are compositional deviations most likely attributed to differences in the nominal vs. measured composition. Because of the large number of samples, the measured compositions were only performed on selected samples and will be presented in a later section.

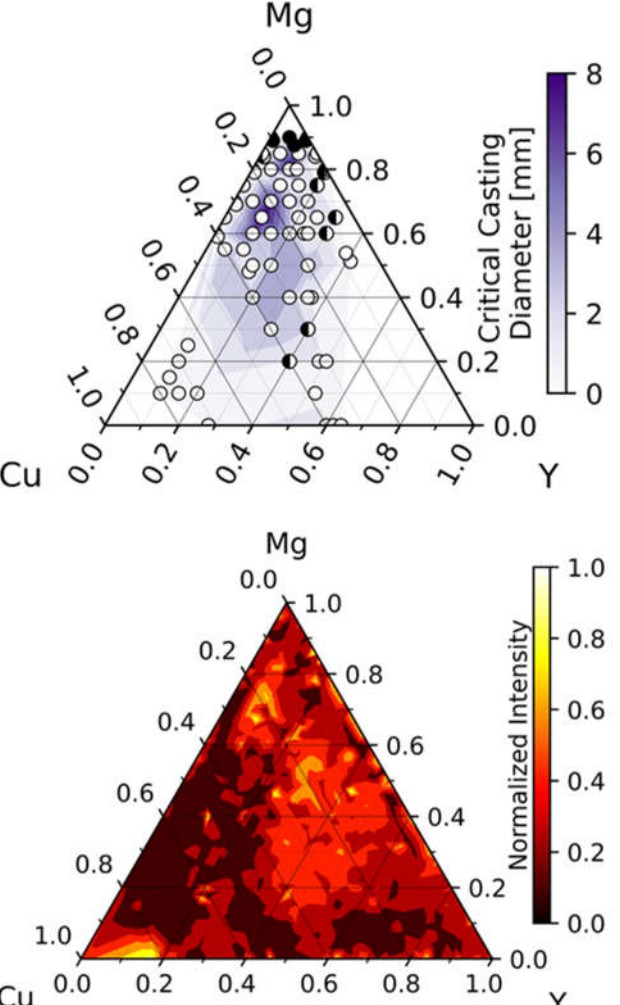

**Figure 5.** (**Top**) a contour map of critical casting diameter (mm) overlaid with a scatterplot showing the experimental literature results for glass-forming ability [61–68] for the given compositions: (○) amorphous, (◐) partly amorphous, and (●) crystalline. (**Bottom**) the normalized optical intensity of the printed beads with total saturation of the camera at 1 and no reflected light at 0. The data are plotted for the nominal composition.

### 3.3. X-ray Diffraction Analysis of High Luminance Samples

The selected XRD results from high luminance beads with measured compositions are shown in Figure 6. There are distinct αMg peaks in addition to several peaks of unknown origin. Since there are a number of complex phases which exist in these compo-

sitional regions, including some with unknown crystal structure, these peaks cannot be identified [69].

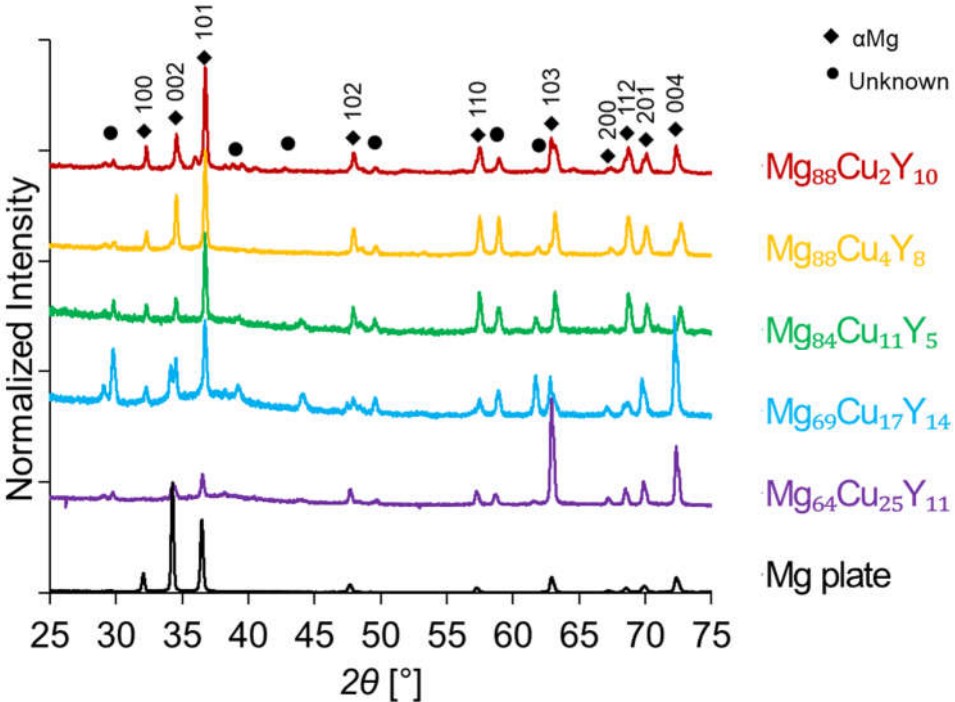

**Figure 6.** Selected XRD traces from the Mg-Cu-Y system with measured compositions and an XRD trace of the Mg substrate for texture reference. Note the broader primary peaks in $Mg_{69}Cu_{17}Y_{14}$ and $Mg_{64}Cu_{25}Y_{11}$. αMg peaks are present in all XRD traces. In the low-Mg samples, the relative peak intensities match those of the powder diffraction file that represent small, randomly oriented grains with no strong texture. The build plate has strong [002] and [201] peaks instead of strong [101] and [112] peaks because the rolled plate has texture. For the low-Mg samples, the relative intensity of the low-angle peaks matches the other beads but the high-angle peaks with disproportionately high intensities match that of the plate. This indicates that there is some phase fraction of αMg at the surface of the bead and that the high-angle peaks are from the substrate and should be discounted when comparing the intensity of the peak broadening to the sharp crystalline peaks. There is another phase unable to be identified with a very small phase fraction; this phase appears in all samples, but with greater signal intensity as the Mg content decreases.

Typically, melt-spun ribbons or suction-cast rods can be analyzed by XRD and be categorized as completely crystalline, partially crystalline, or completely amorphous. The beads can only be categorized as completely crystalline or partially crystalline. The XRD of these beads will always show some crystalline peaks, whether from the crystalline substrate, unmelted powder around the edges of the bead, or crystalline phases beneath the amorphous phase, if present. Because of these crystalline peaks and the non-ideal diffraction conditions of a domed bead, these XRD results must be taken as a rough measure of crystallinity intended only to remove from the sample pool those beads which show nothing but sharp crystalline peaks. A similar principle was applied to the optical tests as well.

Two criteria for the presence of an amorphous phase in the XRD traces of the printed beads were used: amorphous broadening of the XRD pattern and relative peak intensity. Since the Mg plate was used in the as-received condition, it exhibits the texture typical of a rolled plate with higher intensity [002] and [201] peaks rather than [101] and [112] peaks found in typical HCP powder diffraction. In the Mg-rich beads, the relative intensity of the Mg peaks matches the powder diffraction file, indicating a polycrystalline sample with randomly oriented grains. These XRD traces show no indication of peak broadening. In

the lower Mg samples, the relative peak intensity at high values of 2$\theta$ matches the texture of the Mg substrate, but the signal intensity is disproportionate with the low-angle peaks. At lower angles, the X-ray beam is well-aligned with the surface of the bead to ensure that the 35–40° range where the broad amorphous peak for Mg would appear is accurately captured. This causes the beam to be misaligned at higher angles on account of the rounded surface, so the signal comes from the bead at low angles and plate at high angles. Since the signal from the amorphous phase is much smaller than the crystalline signal, the higher angle peaks have a much stronger signal. The amorphous broadening is slight, but coupled with this change in relative intensity, it prevents the sample from being screened out as entirely crystalline.

From the HT synthesis, optical luminance screening, and XRD, two samples appeared to have potential for having amorphous regions. Specifically, the measured compositions (described in the following section) of $Mg_{69}Cu_{17}Y_{14}$ and $Mg_{64}Cu_{25}Y_{11}$ samples were categorized as not completely crystalline and the rest as entirely crystalline.

### 3.4. EDS-SEM Analysis of High Luminance Samples and XRD Screening

A subset of printed beads with higher luminance and with the screening XRD process was analyzed by EDS and the results are shown in Figure 7. The arrows point from the nominal composition towards the measured composition. In some cases, large compositional differences between the nominal and actual composition were determined. The measured compositions cluster around the ternary eutectics in the Mg-rich corner. This is the lowest melting temperature composition in the ternary system. The composition tends to shift towards the Mg-rich corner and away from the Y-rich corner, which would push the high-intensity areas towards the known bulk-glass-forming region. Prior studies of bulk-glass formation in the Mg-Cu-Y system have been concentrated along the 10–15 at% Y range [70].

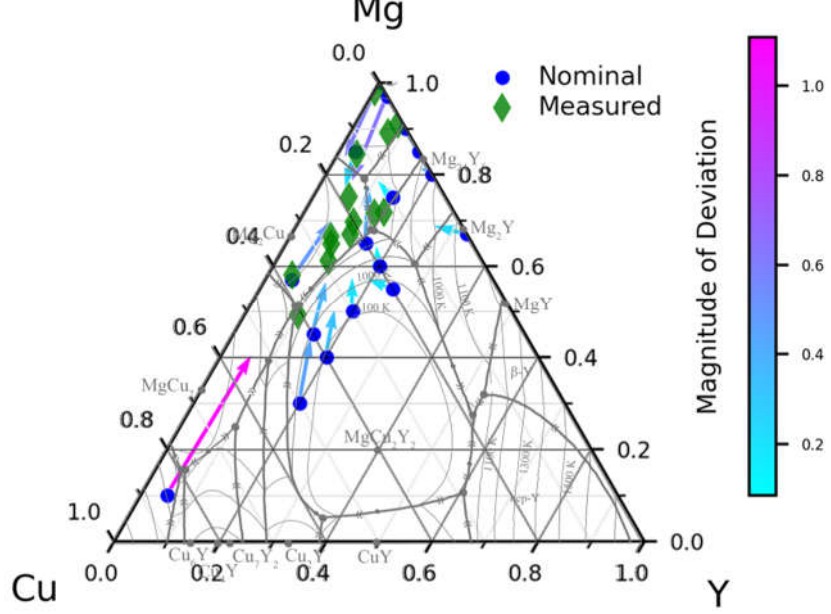

**Figure 7.** The nominal and measured compositions of several printed beads. It is lightly overlaid with the liquidus projection [69,71]. The color and length of the arrows denote the magnitude of the deviation from the nominal composition; this is calculated by taking the square root of the sum of the squares of the difference in atomic fraction for each element. Arrows point from the nominal composition towards the measured composition. The composition generally deviates from very high Mg compositions to slightly lower Mg content and from lower Mg content to slightly higher Mg content. The Y content tends towards the 5–10 at% Y region.

### 3.5. TEM Validation

Based upon the two potential compositions suggesting a potential amorphous nature in XRD, a TEM sample of the $Mg_{64}Cu_{25}Y_{11}$ bead was removed via a focused ion beam (FIB) on a Zeiss Auriga FIB/SEM to explore the amorphous regions; the electron diffraction results can be seen in Figure 8. A lack of crystalline features in the image and the presence of an amorphous halo in the diffraction pattern strongly suggest the presence of an amorphous phase on the surface of this sample. The measured composition via EDS in the TEM of this exact area was $Mg_{54}Cu_{40}Y_6$, which is less Mg than the SEM analysis revealed for the whole, curved sample. The measured composition is consistent with the literature findings that show the best glass-forming region in the system to be in the 5–10 at% Y region [70].

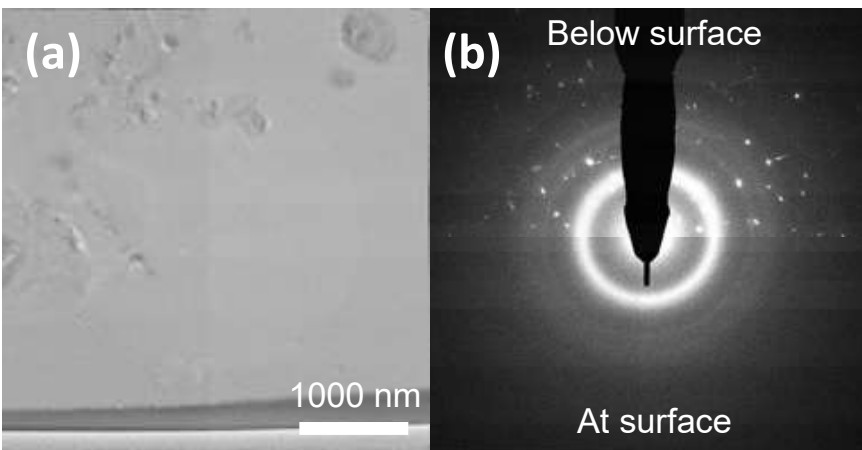

**Figure 8.** (**a**) A TEM image and (**b**) diffraction result taken from a FIB lift-out of the surface of the $Mg_{64}Cu_{25}Y_{11}$ sample. At the very surface of the sample, a distinct lack of features in the image and an amorphous halo was observed. The composition in this spot was measured as $Mg_{54}Cu_{40}Y_6$.

## 4. Discussion

The experimental results in this investigation demonstrate the ability to synthesize 1000 alloy composition samples that can be down-selected to an alloy that has BMG potential in a short period of time. Since the chosen alloy system, Mg-Cu-Y, was a model system for demonstration of the methodology, the amorphous alloy composition determined from the down-selection had been demonstrated as a BMG in prior investigations. The total experimental time of synthesis and characterization was one week, with the TEM investigation being the longest step. The HT methodology and down-selection indicate a potentially successful strategy to explore new alloy systems. However, the coarse screening process also indicates possible deficiencies. Materials discovery with high-throughput (HT) experimental methods can be challenging with competitions between precision and data fidelity. Fundamental information can be lost, or lower quality data may be generated, at the expense of producing more data. The synthesis of bulk samples in large compositional space introduces the need for characterization methods to match the sample production rate. The benefits of the combinatorial experimentation will be discussed, followed by a critical assessment of the data generation.

### 4.1. Combinatorial Experimentation: HT Screening and Down-Selection

Conventional synthesis techniques such as arc casting or melt spinning are limited to a few samples per day, whereas the HT method using DED technology can generate 1000 samples in a day for a three-order-of-magnitude increase in bulk sample production. The technique offers the advantages that "bulk samples" are produced, high cooling rates ($>10^3$ K/s) are achieved, and samples with discrete compositions can be produced. Although the samples are too small for the testing of physical properties, as a synthesis tool for screening GFA, the technique demonstrated the ability to quickly find an amorphous

alloy in a ternary system. As a comparison, thin film depositions can also screen large compositional spaces at rates, but the small size, elastic strains, and continuous composition ranges may yield different results as compared to bulk samples.

With the ability to form bulk samples at high rates, the technique serves better as a screening tool as opposed to a methodology for large-scale sample testing. The process does lend itself favorably to building larger size samples. In the DED process, building a 3D part occurs a layer at a time. The cooling rate will remain sufficiently high to permit amorphous phase formation, but the heat from the deposition of a new layer will cause some degree of crystallization. Therefore, the process is better suited to single tracks or single layers or beads where the deposition spacing is such that heat-affected regions are minimized.

A methodology to synthesize a large dataset of discrete composition alloys introduces the challenge to characterize all of the samples. Clearly, a thorough characterization of each sample can be rate-limiting, regardless of what synthesis technique is used. Alternatively, the optical luminance method was used to down-select potential glass-forming alloys in a rapid manner. While not a definitive test, the method was used as a basis to interrogate samples that may have a more optically reflective signature, potentially indicating a lack of surface roughness. BMGs usually have a "mirror-like" smooth surface characteristic of a liquid. While this optical method could miss potentially amorphous samples, the optical luminance evaluation only took a few minutes. Oxide formation on the sample surfaces was not expected to alter the luminance results. Since the oxygen content of the printing environment is extremely low, the surface of the substrate is clean, and the powder is stored under airtight conditions, there is little oxygen able to get to the melt pool. The beads selected for further analysis were clean and shiny on top, which means that they were free from the oxide skin that would be present with high $O_2$ content. There is no evidence of oxides in the microstructures. The optical luminance characterization could quickly quantify the samples from 1000 (on two plates) to approximately 10–100 samples in this particular investigation.

The XRD and EDS processes take a longer amount of time than the synthesis or optical luminance measurements, depending upon the number of samples chosen to investigate and the amount of scan time used. However, the same time limitations on these methods exist for arc-cast or melt-spun samples. Approximately 10 specimens covering the highest luminance areas were used for the XRD and EDS. Both investigations took approximately a day to collect and analyze the results. The sample surfaces were curved and not expected to yield rigorous measurements. The FIB lift-out and TEM investigations took a day each, respectively, for the single sample. The DED process and analysis provided an HT methodology for synthesis, and down-selection had a tiered level of time commitment. The entire process took five days, or a typical work week. The necessary steps required for the same type of characterization (without luminance) on other conventional bulk techniques such as arc casting or melt spinning were comparable for only a few compositions in the same period of time.

### 4.2. Data Fidelity

At the highest level, and for the intent of this investigation, the goal was to survey a whole ternary system and evaluate whether a BMG alloy could be identified. The model Mg-Cu-Y system was used, and a previously identified composition was found using the HT methodology. While the ability to generate a lot of samples and down-select in a short period of time is useful, the quality of data needs a critical assessment. Most notably, compositional control during synthesis and the characterization methods should be considered. The actual vs. nominal composition were different (Figure 7), and even non-uniform composition distributions were found locally within each bead (as noted in Figure 8).

Compositional control using in situ alloying during DED is affected by the elemental powder or feedstock being used. The control of the mass flow rate of each powder can be

calibrated, but relative incorporation of each element into the melt pool is less controlled. The laser heats the substrate at a single point, creating a temperature gradient. Since the substrate is a pure metal, the melt pool is the area over the melting temperature of Mg with no 'mushy zone'. The powder is injected into the melt pool by the flowing argon, where it melts or reacts with the molten metal to alloy within it. The temperature of the melt pool is limited by the boiling point of the metal, which changes as the alloying content changes. Because the melting temperature of Y is far above the maximum temperature for a pure Mg melt pool, the powder does not melt and reactions are slow, while the melting temperature of Cu is closer to that for Mg, allowing for faster reaction times and faster alloying to increase the temperature of the melt pool, even if it is below the melting temperature of Cu. Thus, there are some compositional regions which do not alloy well, including very low Cu content and high Y content.

The observed amorphous compositions are in the 5–10 at% Y range. This complements the concept that glasses form at compositions with a strong enthalpy of mixing, one of Inoue's empirical rules for glass formation [5]. The compositions which are observed are also more likely to be glassy compositions, even with some compositional deviation. The re-melting of each bead helped increase homogeneity as well as provide a consistent thermal history for each sample.

Another compositional control factor is the baseplate. Mixing with the Mg substrate causes Mg enrichment in the beads, particularly as the melt pool depth increases. The composition deviation shown in Figure 7 demonstrates how the sample compositions deviate from the nominal composition. The arrows point from the nominal composition towards the measured composition, and the color represents the magnitude of the deviation. The composition deviations can be generally attributed to three factors: Mg enrichment from mixing with the substrate, Mg loss from volatilization, and Y enrichment due to its tendency to adhere to the surface of the beads and become incorporated upon re-melting.

The compositional deviation shows potential for the beneficial self-selection of glass-forming compositions. If the cooling rate is insufficient to undercool the liquid below $T_g$ for a specific nominal composition, the phase with the highest driving force for solidification should start to form. This phase should grow until the critical cooling rate of the compositionally deviated remaining liquid is below that of the sample's cooling rate. Then, the remaining undercooled liquid solidifies in an amorphous structure once it reaches $T_g$. For the amorphous beads in this study, that primary growth phase would be αMg and potentially Y-rich phases which serve as nucleation points. The overall composition of the bead lies near the border between the crystalline and amorphous regions, but the calculated composition of the amorphous matrix phase is in the bulk-glass-forming region.

The characterization techniques were chosen to match the synthesis rates as closely as possible. The characterization methodology was designed as a progressive, tiered series of more detailed and time-consuming techniques on fewer samples with higher confidence in the amorphous character. The initial optical technique relied on the smooth and highly reflective surface that amorphous materials are known to have due to the lack of crystalline facets on the surface. The subsequent XRD and SEM-EDS techniques are more time consuming, so they were carried out on a smaller subset of samples. The samples had smaller volumes and were curved surfaces, potentially leading to sampling and accuracy errors of the relative phases and compositions. For example, the Mg substrate was discernable in the sample XRD traces. However, the distinct amorphous broadening near the primary XRD peaks was consistent with the lack of crystalline features observed in TEM. Overall, the process achieved the goal to find a BMG alloy.

## 5. Conclusions

The study presents a high-throughput experimental strategy for screening potential BMG alloys. A model glass-forming ternary system, Mg-Cu-Y, was used to evaluate the process. The methodology demonstrated:

- A total of 1000 "bulk" samples with different compositions can be synthesized in a day.

- A tiered level of HT "screening" permits a down-selection to one successful alloy within a week. The screening included optical luminance, XRD, EDS-SEM analysis, and a final validation of an amorphous composition with a TEM investigation.
- The directed energy deposition synthesis process is three orders of magnitude faster than conventional bulk methods such as arc casting or melt spinning.

The HT method represents a competition between speed and fidelity. The methodology permits successful screening of a large compositional space in a short timeframe. However, the inconsistent composition control and the rapid down-selection method may potentially lead to missing some viable amorphous alloys.

**Author Contributions:** Conceptualization, D.J.T. and J.H.P.; methodology, D.J.T. and J.T.S.; validation, C.S.F.; formal analysis, J.T.S. and C.S.F.; investigation, J.T.S. and C.S.F.; writing—original draft, D.J.T. and J.T.S.; writing—review and editing, J.H.P.; supervision, D.J.T., P.M.V. and J.H.P.; funding acquisition, D.J.T., P.M.V. and J.H.P. All authors have read and agreed to the published version of the manuscript.

**Funding:** This work was supported by the National Science Foundation through the Designing Materials to Revolutionize and Engineer our Future (DMREF) program (Grant #1728933). The electron microscopy was carried out using facilities and instrumentation that are partially supported by the National Science Foundation through the Materials Research Science and Engineering Center (Grant #1720415).

**Data Availability Statement:** The data used to support the findings of this study are available from the corresponding author upon request.

**Conflicts of Interest:** The authors declare no conflict of interest.

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
