# Peer review of "High-Throughput Synthesis and Characterization Screening of Mg-Cu-Y Metallic Glass"

_metals, doi:10.3390/met13071317_

Round 1

Reviewer 1 Report

1. It would be good if the background and purpose of the study were presented in more detail in the abstract.

2. It would be better if the purpose of this study was more clearly stated in the introduction.

3. For all figures in the manuscript, only the title should be summarized so that it is possible to know which figure it is.

And additional explanations about figures should be written in the text of the manuscript.

Correction is required for all figures.

4. The composition from the results of TEM diffraction of Mg64Cu25Y11 was found to be Mg54Cu40Y6.

What is the reason for such a difference in composition?

5. The present conclusion seems to be somewhat insufficient as the final conclusion of this study.

It is thought that it is necessary to introduce a conclusion that can further summarize the results of HT method.

In addition, if the conclusion is supplemented, the abstract using it should be revised together.

Author Response

Reviewer 1

  1. It would be good if the background and purpose of the study were presented in more detail in the abstract.

The abstract does have a limited amount of words to convey the point of the paper. The author’s instructions request no more than 200 words, and our abstract was 187 words.  We try to include the following elements in our abstract, per the author instructions, and a short synopsis is given:

  • Motivation – define a high-throughput experimental methods for BMGs
  • Objective – explore a new technique on a well-studied system
  • Methods – Use in situ alloying with DED and characterization techniques
  • Results – 1000 samples can be screened in a week
  • Conclusion – a BMG was “discovered”, consistent with the literature, 3 orders of magnitude faster than conventional methods.

We view Background, in this context to include the motivation and objective in terms of a broad context and specific purpose.  To convey more of the why (background) and ultimate motives (higher rates of discovery) would require too much additional space, so this was left to the introduction section. In the spirit of the reviewer comment, we have added the following introductory statement to the abstract, “Bulk metallic glasses can exhibit novel material properties for engineering scale components, but the experimental discovery of new alloy compositions is time intensive and thwarts the rate of discovery.

This does put the word count over 200 and the editor can comment on whether this is acceptable.

We apologize if this is missing the reviewer’s point, but the purpose of this study was straight-forward (albeit simple,  it was not easy), and the purpose was clearly stated in the title and first sentence of the abstract. Our intent is to expand this procedure into other potential alloy systems that have different technological impacts.

  1. It would be better if the purpose of this study was more clearly stated in the introduction.

We do whole-heartedly agree with the Reviewer 1 that the Introduction should not be a mystery novel with the “punchline” at the end of the Introduction.  We originally felt that the title and abstract conveyed our goal and purpose very clearly, but we do agree that each section should stand alone. We also believe it should not have extraneous information and get straight to the point. We must be clear that this manuscript reflects a discovery methodology applied to an arguably difficult model system and is not designed for a specific engineering application at this stage of study.  Nonetheless, we have modified the manuscript in the first paragraph of the introduction to be very direct as to the purpose of the effort, by adding, “However, discovery of new glass-forming systems in bulk samples is often limited by the fabrication rate of appropriate samples, and high-throughput screening methods are needed that could accelerate the discovery of new compositions.  This current study focuses on an accelerated discovery method for Mg bulk metallic glasses.” 

  1. For all figures in the manuscript, only the title should be summarized so that it is possible to know which figure it is.

And additional explanations about figures should be written in the text of the manuscript.

Correction is required for all figures.

Internally, we debated this same point many times, and the reviewer makes the point that we considered.  One of the advantages of submitting to Metals is the ability to utilize a free format for the manuscript (per the author’s guide).  As an open access journal, we felt that the technique presented exceeds the rate of other high-throughput experimental methods and would be cited accordingly.  We did not want the manuscript to be difficult to read, with many details saved to the figure captions.  The reviewer is correct that this is NOT what we would normally do in a manuscript, but felt the intended audience would find value in the approach. This is a style feature and decision, and not a content feature as described with earlier comments, so we respectfully request that we maintain the figures “as is”, with our intent explained.  The editor may weigh in on this concept, but we do appreciate the discussion and the ability to tailor the presentation to intended audiences.

  1. The composition from the results of TEM diffraction of Mg64Cu25Y11 was found to be Mg54Cu40Y6. What is the reason for such a difference in composition?

We apologize for the lack of clarity.  While the high-throughput technique allows rapid screening, there are deficiencies that we honestly tried to describe in the Discussion.  However, we did not clearly link this back to Figure 8, so we made this clear in the Discussion with the following statement, “The actual vs. nominal composition were different (Figure 7), and even non-uniform composition distributions were found locally within each bead (as noted in Figure 8).  We also highlighted another statement, “The sample surfaces were curved and not expected to yield rigorous measurements” to reflect our concern over EDS measurements on curved surfaces.  The compositional control turned out to be a much larger effort than the scope of this manuscript.  In the Discussion, we tried to clearly state the deficiencies and compositional competitions between Mg enrichment from the baseplate, Mg volatilization, the lack of Y dissolution (because of Mg volatilization), etc.  We are finding this to be a complex competition where the rates of compositional change are a strong function of process parameters, feedstock, and thermophysical properties.  Research is in progress, but this does not negate the usefulness of the screening method as presented!

  1. The present conclusion seems to be somewhat insufficient as the final conclusion of this study.

It is thought that it is necessary to introduce a conclusion that can further summarize the results of HT method.

In addition, if the conclusion is supplemented, the abstract using it should be revised together.

We tried to follow the Authors Guide…

Metals does not require a Conclusion, but the Guide to Authors state that a Conclusion can be added to simplify/condense the Discussion section. That is what we did in the Conclusion. Per the author’s guide recommendation, therefore, we focused on our primary objective to increase experimental discovery rate, and the conclusion clearly states our finding as this final point. We felt that we spent considerable time critically evaluating potential deficiencies in the Discussion, and we wanted readers to recognize that despite competitions between rate and fidelity, the rate of discovery in an entire ternary system was distinctly higher than other BMG fabrication and screening techniques.  We are glad to omit this section at the editor’s discretion, but we do feel the section succinctly reduces a large amount of data to a useful “takeway”.

Reviewer 2 Report

Authors presented in the manuscript a new method for the high throughput screening of bulk metallic glasses. The method naturally derives from previous literature trials with general alloy screening and BGA single composition production through directed energy deposition (DED) method. The data contained in the manuscript present valuable input in the field and demonstrate a good potential for future in-depth research. However, there are few shortcomings that deserve more attention for necessary improvements:

-Newer references from 2020 onwards were not presented in the manuscript Papers are found in the literature at a simple search, for example treating alloy screening by DED or BMG production by DED. A review on BMGs is also found.

- While the optical and electron microscopy was specified in discussed in the manuscript, there are no images to show the referred assumptions

- Conclusions are mentioned as a result statement and in reality should present the findings and not the achievements (results). The deficiencies need to be also presented.  Considerable improvement is required.

Author Response

Reviewer 2

Authors presented in the manuscript a new method for the high throughput screening of bulk metallic glasses. The method naturally derives from previous literature trials with general alloy screening and BGA single composition production through directed energy deposition (DED) method. The data contained in the manuscript present valuable input in the field and demonstrate a good potential for future in-depth research.

We appreciate this comment and can assure the reviewer that the future in-depth research is in progress.  This is a new area and is growing!

However, there are few shortcomings that deserve more attention for necessary improvements:

-Newer references from 2020 onwards were not presented in the manuscript. Papers are found in the literature at a simple search, for example treating alloy screening by DED or BMG production by DED. A review on BMGs is also found.

This was a slight oversight on our part because the refinement of this research took longer than expected and out-lived the original literature review.  We have modified and highlighted additional references. This does not alter the impact or novelty of the research, but it does provide more context for the need of these types of high-throughput data. The older references are maintained to demonstrate where we felt original concepts were first presented.  In addition, the added references include different materials and recent reviews. Specifically, we added the following references:

  • [11] G. Liu, S. Sohn, Sebastian A. Kube, A. Raj, A. Mertza, A. Nawano, A. Gilbert, M.D. Shattuck, C.S. O’Hern, J. Schroers, Machine learning versus human learning in predicting glass-forming ability of metallic glasses, Acta Materialia 2023 243 1189497.

(page 2, line 55)

  • [46] W. Wu, X. Li, Q Liu, J. Ying Hsi Fuh, A. Zheng, Y. Zhou, L. Ren, G. Li, Additive manufacturing of bulk metallic glass: Principles, materials and prospects, Materials Today Advances 2022 16 100319.

(page 2, Line 86)

  • [47] N. Sohrabi, J. Jhabvala, R.E. Logé, Additive manufacturing of bulk metallic glasses—process, challenges and properties: A Review, Metals 2021 11 1279.

The authors are open to any additional papers that the reviewers deem noteworthy!  The reference numbering was adjusted accordingly.

- While the optical and electron microscopy was specified in discussed in the manuscript, there

are no images to show the referred assumptions

This comment is a bit unclear to us, so we will try to clarify.  Optical methods refer to the reflectivity and luminance methods.  Optical micrographs were not presented other than the macro-image of Figure 4.  SEM was used for Figure 1, and the only other SEM work presented were the EDS measurements.  The only microstructure shown was a TEM micrograph, and one was shown in Figure 8.

Originally, we did have micrographs but found them to not add content or support conclusions, so we did eliminate the micrographs.  If a remnant discussion still exists, it would help us if the page and line numbers were specified.  We did check and think that this was not the case, but we value any comments that suggest otherwise!  Sorry for the confusion, but we think we were clear?

- Conclusions are mentioned as a result statement and in reality should present the findings and

not the achievements (results). The deficiencies need to be also presented. Considerable

improvement is required.

A similar comment came from Reviewer 1.  We tried to follow the Authors Guide. We repeat the response to Reviewer 1 below…

Metals does not require a Conclusion, but the Guide to Authors state that a Conclusion can be added to simplify/condense the Discussion section. That is what we did in the Conclusion. Per the author’s guide recommendation, therefore, we focused on our primary objective to increase experimental discovery rate, and the conclusion clearly states our finding as this final point. We felt that we spent considerable time critically evaluating potential deficiencies in the Discussion, and we wanted readers to recognize that despite competitions between rate and fidelity, the rate of discovery in an entire ternary system was distinctly higher than other BMG fabrication and screening techniques.  We are glad to omit this section at the editor’s discretion, but we do feel the section succinctly reduces a large amount of data to a useful “takeway”.

Reviewer 3 Report

D.J. Thoma et al studied high-throughput synthesis and characterization screening of Mg-Cu-Y metallic glass. The work shows interesting routes and results. It can be considered for acceptance after the following revisions. 

1. Is this high throughput synthesis method universal? Are there limitations to this process for raw powders with different melting points and particle sizes? Please provide some description and discussion on the above issues in the revised manuscript.

2. Does the material have a certain burn rate? Does such result cause the composition shift? Is there oxidation on the surface of the material? What is the impact on the subsequent analysis? Please have a brief discussion.

3. I'm very positive about the high-throughput synthesis in this study. For the rapid screening of amorphous alloy components in 1000 samples with different compositions, the author narrowed the composition range step by step by optical observation, XRD, EDS-SEM, and finally verified it by TEM. The TEM verification step appears to be the most time-consuming and costly. Is there a less time-consuming and less costly verification method?

4. If TEM verification is not performed, how accurate is the prediction of the previous step? Please discuss it.

5. Abstract and introduction should highlight novelty.

6. Fig 8. Please number the multiple subgraphs in the group separately. 

7. Some references are too old. Authors need to state and cite more related works in the last two years. 

Minor editing of English language required.

Author Response

Reviewer 3

  1. Is this high throughput synthesis method universal? Are there limitations to this process for raw powders with different melting points and particle sizes? Please provide some description and discussion on the above issues in the revised manuscript.

We believe the process is universal, but each system of alloys and starting feedstock will introduce their own calibration requirements.  To make a bold statement without evidence is a bit speculative and too premature of a discussion at this time.  We did include a complete section in the Discussion on issues with melting point, volatilization, and baseplate dissolution. We are confident that additional issues can be factors with different alloy systems, but that would be pure hypotheses until the studies were completed.  We did opt to rely only on the data at hand, but we are working on new systems and publications.  The Mg-Cu-Y system did offer one of the more difficult systems owing to the high melting temperature of Y as compared to the volatilization temperature of Mg, and this is described in the Discussion section.  We think your comments are very correct and addressed them in the manuscript, but we did not want to add speculative comments without further data.

  1. Does the material have a certain burn rate? Does such result cause the composition shift? Is there oxidation on the surface of the material? What is the impact on the subsequent analysis? Please have a brief discussion.

We do not believe oxidation is an issue owing to the low amount of oxygen. This is stated in the manuscript.  In fact, as a side note, we must carefully and slowly expose the system to oxygen when opening the glove box for cleaning.  The residual particulate and volatilization product can smolder when exposed to air, suggesting oxidation. 

While the high-throughput technique allows rapid screening, there are deficiencies that we honestly tried to describe in the Discussion.  The compositional control turned out to be a much larger effort than the scope of this manuscript.  In the Discussion, we tried to clearly state the deficiencies and compositional competitions between Mg enrichment from the baseplate, Mg volatilization, the lack of Y dissolution (because of Mg volatilization), etc.  We are finding this to be a complex competition where the rates of compositional change are a strong function of process parameters, feedstock, and thermophysical properties.  Research is in progress, but this does not negate the usefulness of the screening method as presented! We do think that volatilization and baseplate dissolution are countering effects in the Mg content that also affects Y capture efficiency.  Trying to predict this is much larger than the scope of this study and is currently underway.  This is an astute observation by the reviewer, but any discussion at this point (beyond what is stated) is only a speculation as this point and does not provide value.  Despite these noted deficiencies, the process was effective for screening an alloy.

  1. I'm very positive about the high-throughput synthesis in this study. For the rapid screening of amorphous alloy components in 1000 samples with different compositions, the author narrowed the composition range step by step by optical observation, XRD, EDS-SEM, and finally verified it by TEM. The TEM verification step appears to be the most time-consuming and costly. Is there a less time-consuming and less costly verification method?

Again, the reviewer’s observations are correct.  TEM is the rate limiting step.  The balance between high-throughput screening and data fidelity is an issue that we are actively pursuing, but for this study, only TEM would convince us, despite the apparent peak broadening in XRD.  TEM confirmation and validation is quite common for BMGs in the literature.  We have added the following comment on page 7, line 210, “TEM validation was considered as the only definitive confirmation of an amorphous phase.”

  1. If TEM verification is not performed, how accurate is the prediction of the previous step?Please discuss it.

We respectfully believe this to be a moot point.  None of the authors believed a credible argument could be made on the existence of an amorphous phase without TEM validation.

  1. Abstract and introduction should highlight novelty.

Many journals specifically state novelty should be apparent without saying “new”, “novel”, “breakthrough”, etc.  However, we do agree with reviewer that this does assist the reader.  Please note in the manuscript we have highlighted two sections.  In the abstract, we have the statement “The tiered rate of amorphous alloy synthesis and characterization can survey a large compositional space and permits a glass-forming range to be identified within one week, making the process at least three orders of magnitude faster than other discrete composition techniques such as arc-melting or melt-spinning.”  We believe this statement punctuates the novelty of the research!  We have also added the clarifying purpose in the abstract, “Bulk metallic glasses can exhibit novel material properties for engineering scale components, but the experimental discovery of new alloy compositions is time intensive and thwarts the rate of discovery. “  This statement defines the need for a novel experimental methodology to screen and discover new BMGs.  Finally, we have added the statement, “However, discovery of new glass-forming systems in bulk samples is often limited by the fabrication rate of appropriate samples, and high-throughput screening methods are needed that could accelerate the discovery of new compositions.  This current study focuses on an accelerated discovery method for Mg bulk metallic glasses.” into the introduction to highlight the novel solution this manuscript provides for accelerated materials discover.

  1. Fig 8. Please number the multiple subgraphs in the group separately.

Thank you.  We missed that with a last-minute change to the Figure 8!  Fixed and highlighted!

  1. Some references are too old. Authors need to state and cite more related works in the last two years.

This was a slight oversight on our part because the refinement of this research took longer than expected and out-lived the original literature review.  We have modified and highlighted additional references. This does not alter the impact or novelty of the research, but it does provide more context for the need of these types of high-throughput data. The older references are maintained to demonstrate where we felt original concepts were first presented.  In addition, the added references include different materials and recent reviews. Specifically, we added the following references:

  • [11] G. Liu, S. Sohn, Sebastian A. Kube, A. Raj, A. Mertza, A. Nawano, A. Gilbert, M.D. Shattuck, C.S. O’Hern, J. Schroers, Machine learning versus human learning in predicting glass-forming ability of metallic glasses, Acta Materialia 2023 243 1189497.

  • [46] W. Wu, X. Li, Q Liu, J. Ying Hsi Fuh, A. Zheng, Y. Zhou, L. Ren, G. Li, Additive manufacturing of bulk metallic glass: Principles, materials and prospects, Materials Today Advances 2022 16 100319.

  • [47] N. Sohrabi, J. Jhabvala, R.E. Logé, Additive manufacturing of bulk metallic glasses—process, challenges and properties: A Review, Metals 2021 11 1279.

The authors are open to any additional papers that the reviewers deem noteworthy!  The reference numbering was adjusted accordingly.

Finally, Reviewer 3 suggested minor editing is required.  We did have the paper thoroughly reviewed by internal sources before submission, so we are confused as to what grammatical deficiencies exist without a bit more detail.  We do acknowledge that many mistakes were made in the translation of our submission to the formatted version sent to reviewers.  For example, many sub-scripts and super-scripts were not properly typeset from our submitted version.  We did correct and highlight these mistakes in the revised manuscript.  Please advise if additional issues exist, as this was not a consensus observation!  We are dedicated to ensure a readable document!

Reviewer 4 Report

The paper "High-Throughput Synthesis and Characterization Screening of Mg-Cu-Y Metallic Glass" is a very well written report on high throughput screening of the compositional space using an additive manufacturing technique.

The approach is noval and very promising. The great quality of the manuscript is reflected by the acknowledment of the authors that at the experimental reality is often more complex than the initial concept might suggest. The authors very clearly identify the problems of the chosen approach, particularly in combining it with further characterization techniques.

One technique that could be very interesting is using polarized light or other optical filtering (how about dark field optical microscopy?) to identify the clearly crystalline beads.

In general this is a good and interesting report and will be very useful to the community. The paper can be accepted as is.

Author Response

Reviewer 4

We would like to thank the reviewer for the positive feedback. We do believe this technique has potential for new materials discovery.  We continue to work on the compromise and competition between experimental rate and data fidelity.

Round 2

Reviewer 1 Report

The authors' responses were well reviewed.

The efforts of the authors to revise the manuscript are visible.

I totally agree with the author's opinion.

I think the role of a reviewer is to help the submitted paper be completed in a more qualitative way, unless there is a separate problem.

However, I feel that the intention of the reviewer is not conveyed well about this review.

- I am aware that there is a limit to the number of words that can be used in an abstract. However, as the summary represents the characteristics of this thesis, the authors need to write it carefully. From that point of view, it was thought that it would be nice if the abstract appeared more clearly such as background, purpose etc.

- It is not a request from the reviewer to ask the authors to conform to the style of the paper.

It is natural that the opinions of the authors should be written freely in the thesis.

However, I think published papers should be easy for many readers to read and understand.

In that respect, it is requested that the manuscript be prepared.

The current manuscript is difficult to read due to scattered content.

- Also, the conclusion represents the final opinion on this paper.

It is judged that readers should be interested in this thesis and understand the conclusion.

In that sense, wouldn't it be helpful to improve this thesis if the conclusion of the content related to the HT method, which is the point of this thesis, is mentioned more intensely?

Considering the above opinions, it is judged that the thesis will be further improved.

I think it would be good for the authors to consider.

Author Response

The authors' responses were well reviewed.

The efforts of the authors to revise the manuscript are visible.

I totally agree with the author's opinion.

I think the role of a reviewer is to help the submitted paper be completed in a more qualitative way, unless there is a separate problem.

However, I feel that the intention of the reviewer is not conveyed well about this review.

- I am aware that there is a limit to the number of words that can be used in an abstract. However, as the summary represents the characteristics of this thesis, the authors need to write it carefully. From that point of view, it was thought that it would be nice if the abstract appeared more clearly such as background, purpose etc.

We do appreciate the reviewer’s comments and take the feedback very seriously.  We do apologize if there was a misunderstanding, and we did add an introductory statement to the abstract to accommodate the request.  Thank you for the clarification!

- It is not a request from the reviewer to ask the authors to conform to the style of the paper.

It is natural that the opinions of the authors should be written freely in the thesis.

However, I think published papers should be easy for many readers to read and understand.

In that respect, it is requested that the manuscript be prepared.

The current manuscript is difficult to read due to scattered content.

We fully acknowledge the reviewer’s intent on this comment, and we have thought this through from a variety of perspectives.  Our conclusion was that this paper presents a significant leap in high-thoughput screening for a variety of audiences, not just the amorphous materials community.  As such, we tried to limit too much descriptive detail to the caption, a strategy that we normally would not do.  With multiple internal repetitions, we felt that a “simple” read amplified the point we were trying to make about the manuscript while still maintaining the rigor expected for the technical community.  We do expect that the reviewer will be correctly representing many readers, but we do believe that a larger majority will prefer the format we used.  We did check this with a large test group within our own organization for feedback, and therefore, since the technical content is not diminished, we used the as-submitted style.

- Also, the conclusion represents the final opinion on this paper.

It is judged that readers should be interested in this thesis and understand the conclusion.

In that sense, wouldn't it be helpful to improve this thesis if the conclusion of the content related to the HT method, which is the point of this thesis, is mentioned more intensely?

We did alter the conclusion in this round, per the reviewers suggestion.  The conclusion now reads:

The study presents a high-throughput experimental strategy for screening potential BMG alloys.  A model glass forming ternary system, Mg-Cu-Y, was used to evaluate the process.  The methodology demonstrated:

  • 1000 “bulk” samples with different compositions can be synthesized in a day.
  • A tiered level of HT “screening” permits a down-selection to one successful alloy within a week. The screening included optical luminance, XRD, EDS-SEM analysis, and a final validation of an amorphous composition with a TEM investigation.
  • the directed energy deposition synthesis process is three orders of magnitude faster than conventional bulk methods such as arc casting or melt spinning.

The HT method represents a competition between speed and fidelity. The methodology permits successful screening of a large compositional space in a short timeframe.  However, the inconsistent composition control and the rapid down-selection method may potentially lead to missing some viable amorphous alloys.

Considering the above opinions, it is judged that the thesis will be further improved.

I think it would be good for the authors to consider.